# Infusing Spatial Knowledge into Deep Learning for Earth Science: A Hydrological Application

**Zelin Xu**
Department of CISE
University of Florida
Gainesville, Florida, USA
`zelin.xu@ufl.edu`

**Tingsong Xiao**
Department of CISE
University of Florida
Gainesville, Florida, USA
`xiaotingsong@ufl.edu`

**Wenchong He**
Department of CISE
University of Florida
Gainesville, Florida, USA
`whe2@ufl.edu`

**Yu Wang**
Department of MAE
University of Florida
Gainesville, Florida, USA
`yuwang1@ufl.edu`

**Zhe Jiang**
Department of CISE
University of Florida
Gainesville, Florida, USA
`zhe.jiang@ufl.edu`

## Abstract

The integration of Artificial Intelligence (AI) into Earth science, including areas such as geology, ecology, and hydrology, brings potential for significant advancements. Despite this potential, applying deep learning techniques to spatial data in this field is often hindered by the lack of domain knowledge. This paper studies the integration of spatial domain knowledge and deep learning for Earth science. The problem is challenging due to the sparse and noisy input labels, spatial uncertainty, and high computational costs associated with a large number of sample locations. Existing works on neuro-symbolic models focus on integrating symbolic logic into neural networks (e.g., loss function, model architecture, and training label augmentation), but these methods do not fully address the specific spatial data challenges. To bridge this gap, we propose a Spatial Knowledge-Infused Hierarchical Learning (SKI-HL) framework[1], which iteratively infers labels within a multi-resolution hierarchy, and trains the deep learning model with uncertainty-aware multi-instance learning. The evaluation of real-world hydrological datasets demonstrates the enhanced performance of the SKI-HL framework over several baseline methods. The code is available at `https://github.com/ZelinXu2000/SKI-HL`.

## 1 Introduction

In recent years, the potential of Artificial Intelligence (AI) to revolutionize a variety of scientific domains has been widely recognized [25]. Earth science, including disciplines such as geology [13], ecology [7], and hydrology [23, 10], stands as a prominent field where AI brought about transformative change. However, one major bottleneck is the lack of spatial domain knowledge in deep learning models, which is a crucial element in Earth science research. This paper studies the integration of spatial domain knowledge and deep learning for Earth science, focusing on flood mapping as a representative hydrological application.

Given Earth imagery, normally sparse training labels, a base deep neural network model, and a spatial knowledge base with label constraints, our problem is to infer the full labels while training the neural network. For the example of flood mapping on Earth imagery, data samples are Earth imagery pixels

---

[1]This work has been accepted to ACM SIGSPATIAL 2023.

NeurIPS 2023 AI for Science Workshop.

in a raster grid, and explanatory feature layers are spectral bands. Initial noisy labels can be from volunteered geographical information (e.g., geo-tagged tweets). These labels are sparse and limited, as collecting complete high-quality labels through manual annotation is impractical (e.g., high time costs, obscured view due to tree canopies near flood boundary). On the other hand, there exists spatial domain knowledge related to topographical constraints on floodwater distribution, e.g., if location A is flooded and location B is at a nearby lower location, then B is flooded. Similar examples exist in crop type classification [22], tree crown delineation in forest ecology [7]', and land use classification [13].

However, the problem poses unique challenges. First, the input labels are always spatially sparse and noisy, making it difficult to train a neural network on Earth imagery directly. For example, in flood mapping, in-situ water sensors are often located at a few locations. Second, spatial uncertainty is inherent in the knowledge-guided label inference process, which comes from the noise and sparsity of input labels, imperfect knowledge and rules, and grounding spatial rules on a coarse grid. Third, there are high computational costs associated with spatial logic inference on a large number of raster pixels and a trade-off has to be made between computational efficiency and spatial granularity.

The most closely related works are neural-symbolic systems, which integrate symbolic logical reasoning with deep neural networks [6]. Existing methods focus on replacing the search process of symbolic reasoning to neural network [20, 32, 19], convert unstructured data, *e.g.*, images into symbols for relational learning [31], representing symbolic knowledge as a loss regularization term [11, 4, 5, 29, 28, 33, 2], or the combination of logic inference of pseudo-labels and neural network training iteratively [18, 26, 3, 12, 17, 24]. However, these methods do not fully address the inherent challenges of spatial data, such as spatial uncertainty and the substantial computational burdens associated with logical inference over a massive number of samples (pixels).

To address the limitations of existing works, we propose **S**patial **K**nowledge-**I**nfused **H**ierarchical **L**earning (SKI-HL) [30] that integrates deep learning techniques with spatial knowledge-infused label inference [15, 1]. SKI-HL consists of two main modules: the uncertainty-guided hierarchical label inference module and the uncertainty-aware deep learning module. The uncertainty-guided hierarchical label inference module captures spatial relationships and dependencies based on a spatial knowledge base and infers labels with quantified uncertainty. To handle the continuous space issue, we design a multi-resolution hierarchy to iteratively refine labels with a trade-off between granularity and computational efficiency. The uncertainty-aware deep learning module leverages complete but uncertain labels from the label inference module, capturing information from the data features that cannot be obtained through logical reasoning alone. Both modules are trained iteratively to refine inferred labels, reduce uncertainty, and improve deep learning model performance. In summary, the contributions of this paper are as follows:

- We propose SKI-HL, a spatial knowledge-infused framework that integrates deep learning and logical reasoning to leverage both explanatory features and spatial knowledge derived from domain logic rules.

- Our approach is designed to handle uncertainty in both the original labels and the label inference process, making it more robust and reliable.

- We propose a strategy to balance the trade-off between spatial accuracy and computational efficiency when discretizing continuous spatial spaces for constructing logic rules and training deep learning models.

- Taking the flood mapping problem as an example, extensive experiments on real-world datasets demonstrate the superior performance of our model compared to baseline methods.

## 2 Problem statement

### 2.1 Preliminaries

**Spatial Raster Framework:** A spatial raster framework is a tessellation of a two-dimensional plane into a regular grid of $N$ cells. The framework can consist of $m$ non-spatial explanatory *feature layers* and *one class layer*. We denote the explanatory feature layers by $\mathbf{X} = \{\mathbf{x}_1, \mathbf{x}_2, \cdots, \mathbf{x}_N\}$ and the class layer by $\mathbf{Y} = \{y_1, y_2, \cdots, y_N\}$, where $\mathbf{x}_i \in \mathbb{R}^{m \times 1}$ and $y_i$ are the explanatory features, and class at cell $i$ respectively. Each cell in a raster framework is a spatial data sample, note as

Table 1: An example of a spatial knowledge base in flood mapping.

| Spatial Rules |
| --- |
| $\forall s_i, s_j \left( Flood(s_i) \wedge Adjacent(s_i, s_j) \right) \rightarrow Flood(s_j)$ |
| $\forall s_i, s_j \left( River(s_i) \wedge Distance(s_i, s_j) \leq d \right) \rightarrow Flood(s_j)$ |
| $\forall s_i, s_j \left( Flood(s_i) \wedge River(s_i) \wedge Downstream(s_i, s_j) \right) \rightarrow Flood(s_j)$ |
| $\forall s_i \left( LandCover(s_i, \text{Wetlands}) \wedge HeavyRain(s_i) \right) \rightarrow Flood(s_i)$ |
| $\forall s_i \left( Slope(s_i) > s \right) \rightarrow \neg Flood(s_i)$ |
| $\forall s_i \left( Elevation(s_i) > e \right) \rightarrow \neg Flood(s_i)$ |

$\mathbf{s}_i = (\mathbf{x}_i, y_i)$, where $i \in \mathbb{N}, 1 \leq i \leq N$. For example, in the flood mapping problem, the explanatory features are the spectral bands from remote sensing imagery, the target classes are flood and dry categories, and each pixel in the image is a spatial sample.

**Spatial Knowledge Base:** A spatial knowledge base $\mathcal{KB}$ is a set of logic rules: $\mathcal{KB} = \{r_1, r_2, \cdots, r_{|\mathcal{KB}|}\}$. Here, each $r_i$ is a rule that represents a spatial relationship, dependency, or constraint between entities in the set of spatial samples $\mathbf{S}$. The quantity $|\mathcal{KB}|$ represents the number of rules in the spatial knowledge base $\mathcal{KB}$.

Table 1 provides an example of a spatial knowledge base used for a flood mapping on Earth imagery problem. Here the variable $s_i, s_j$ stands for a location in the study area or a pixel of Earth imagery. It is important to clarify that these rules are probabilistic in nature, reflecting the likelihood of a flood occurrence under certain conditions, rather than providing an absolute certainty. Please see Appendix A for more preliminaries about the logic rule.

## 2.2 Problem definition

Formally, we define our problem as follows: given a large-scale spatial raster framework with spatial samples $\mathbf{S} = \{\mathbf{s}_1, \mathbf{s}_2, \cdots, \mathbf{s}_N\}$, a set of explanatory feature layers $\mathbf{X}$ in $\mathbf{S}$, a limited set of labels $\mathbf{Y_1} = \{y_1, y_2, \cdots, y_l\}$, usually $l \ll N$, each label associated with a sample in $\mathbf{S}$, a spatial knowledge base $\mathcal{KB}$, and a base neural network model (e.g., U-Net [21]), the output will be 1) Inferred labels $\hat{\mathbf{Y}}$ with quantified uncertainty $\mathbf{U}$, and 2) A deep learning model $DL : \mathbf{X} \rightarrow \mathbf{Y}$. Our objective is to maximize the consistency between inferred labels $\hat{\mathbf{Y}}$ and $\mathcal{KB}$ and maximize the prediction accuracy of the deep learning model.

Specifically, we assume the raster framework $\mathbf{S}$ contains a large number of pixel samples but only with a limited set of labels $\mathbf{Y_1}$. The main objective is to predict the class layer $\mathbf{Y}$ for all spatial samples. To illustrate, consider the case of flood mapping on Earth imagery. In this scenario, the set of spatial samples $\mathbf{S}$ corresponds to Earth imagery pixels. The explanatory feature layers $\mathbf{X}$ are the spectral bands. The label set $\mathbf{Y}$ corresponds to the flood status of each pixel (*i.e.*, flooded or not). The spatial knowledge contains domain constraints on flood locations (e.g., terrains and topography), which is used to infer flood labels $\hat{\mathbf{Y}}$. Considering the errors in the inference process and the imperfect logic rules, uncertainty $\mathbf{U}$ naturally exists in the inferred label.

# 3 The proposed approach

## 3.1 Overview

Our task is to train the deep learning model and infer sample labels based on the spatial knowledge base. The task is non-trivial for several reasons. First, spatial knowledge inference on labels is computationally expensive due to the immense volume of spatial samples and complex spatial dependencies and interactions. Therefore, scalable grounding strategies are required that can effectively handle these issues by balancing computational efficiency and grounding granularity. In addition, the label inference is complicated due to incomplete and sparse initial labels compared with the large study area. Such a low proportion of known data makes logic inference difficult. Furthermore, the labels inferred come with uncertainty at different granularity levels, which is non-trivial for the training of a deep learning model.

To address these challenges, we propose a **S**patial **K**nowledge-**I**nfused **H**ierarchical **L**earning (SKI-HL) framework. Our SKI-HL framework, illustrated in Figure 1, consists of two interdependent modules. The hierarchical label inference module infers sample labels in the raster framework with a trade-off between computational efficiency and spatial granularity. We formulate the inference process as an optimization problem with an objective based on the distance loss from Probabilistic Soft Logic (PSL) [15, 1], and the spatial grounding configuration in a multi-resolution hierarchical grid structure. We design a greedy heuristic to iteratively refine the inferred labels based on inferred spatial uncertainty. The uncertainty-aware deep learning module trains neural network parameters from uncertain labels in multiple resolutions by an uncertainty-aware loss function and multi-instance learning. The two modules run in iterations: the outputs of the deep learning model will serve as the initialization of the hierarchical label inference module in the next iteration.

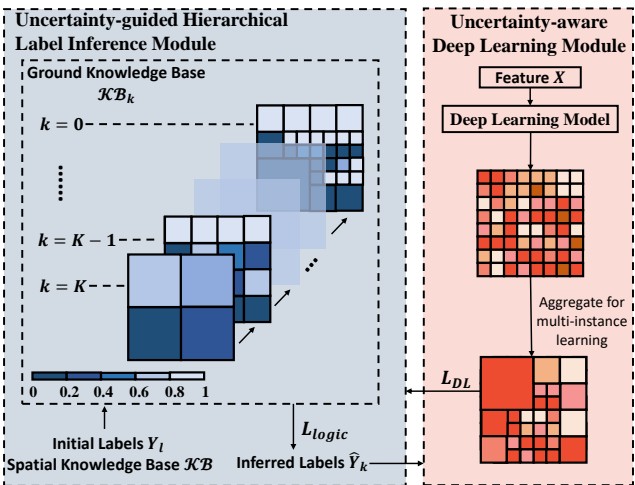

Figure 1: Framework of SKI-HL.

## 3.2 Hierarchical label inference with spatial knowledge

The hierarchical label inference module utilizes spatial knowledge to enhance the label inference process within a raster framework. The module seeks spatial grounding within the framework to optimize sample class probabilities, considering the balance between granularity, computational efficiency, and inference accuracy in handling large-scale spatial data. This approach avoids excessive computational costs associated with high-resolution pixels and the coarseness of low-resolution pixel blocks. The challenge is framed as an optimization problem, focusing on generating accurate sample labels and ensuring a balance in the spatial grounding process.

### 3.2.1 Optimization objective

We now formulate the spatial logic inference of sample labels in a raster framework as an optimization problem. First, we need to define the candidate feasible solution of spatial grounding. The process of spatial grounding refers to substituting the variables in the knowledge base rules with specific, concrete instances, which in our case are spatial samples such as pixels in Earth imagery. Given a set of spatial samples and rules from a spatial knowledge base, we substitute each possible sample into the rules to generate candidate feasible solutions for spatial grounding.

**Spatial hierarchical structure:** We exploit a hierarchical framework to address the hierarchical and fractal pattern of spatial relationships. A large-scale spatial raster framework of pixels is represented at multiple resolutions. At the coarse level, we can treat each cell as a condensed representation of many pixels. In the hierarchical structure of the spatial raster, each cell condenses many pixels, simplifying the initial logic inference stage by reducing the ratio of unlabeled data, making inference more feasible. Please see Appendix B for more details.

Second, we need to define the loss function based on the spatial grounding and inferred label probabilities. To make inferences that are consistent with spatial knowledge, we adopt t-norm fuzzy

logic (see Appendix A) to define the extent of a rule as satisfied, which relaxes binary truth values to a continuous value between $[0, 1]$. Then, following the structure of Probabilistic Soft Logic (PSL), we can induce the distance $d_r(I) = \max\{0, I(r_{body}) - I(r_{head})\}$ to satisfaction for a rule $r : r_{body} \rightarrow r_{head}$, where $I$ is the soft truth value function that can map an atom $a$ or a rule $r$ to an interval between $[0, 1]$, indicating the probability that the atoms or rule holds. PSL determines a rule $r$ as satisfied when the truth value of $I(r_{body}) - I(r_{head}) \geq 0$. To this end, we can convert logical sentences into convex combinations of individual differentiable loss functions, which not only improves the training robustness but also ensures monotonicity with respect to logical entailment, *i.e.*, the smaller the loss, the higher the satisfaction. Therefore, given a set of ground rules in the ground knowledge base $\mathcal{KB}$, we can obtain the truth value for all ground atoms, which can serve as inferred labels for the spatial samples.

$$\hat{\mathbf{Y}} = \arg\min_{I} \sum_{r \in \mathcal{KB}} \omega_r d_r(I) \tag{1}$$

where $\omega_r$ is the weight of rule $r$. It is noted that here the inferred labels $\hat{\mathbf{Y}}$ are not binary values but soft truth values between $[0, 1]$.

Therefore, in our hierarchical framework, our optimization problem can be summarized as searching for an optimal grounding strategy and minimizing the overall distance to satisfaction for the ground atoms. we formally define the objective to minimize in the hierarchical label inference module as:

$$L_{logic} = \sum_{k=1}^{K} \left( \sum_{r \in \mathcal{KB}_k} \omega_r d_r(I_k) + \lambda |\mathcal{KB}_k|) \right) \tag{2}$$

where $\mathcal{KB}_k$ stands for the ground knowledge base in the $k$-th layer, $\lambda$ is a balancing coefficient. The summation over $k$ stands for the overall objective of all layers in the hierarchical structure. The first term is the loss defined by PSL distance, which can drive accurate inference. The second term is used to decrease the ground atoms in each layer.

### 3.2.2 A greedy algorithm

To make a balance between inference accuracy, efficiency, and granularity, we proposed a greedy heuristic grounding strategy. Intuitively, uncertain atom inference always causes a higher distance to the satisfaction of a rule, so here we choose uncertain cells in a coarse layer to refine. The quantified uncertainty $u_{k,i}$ for each cell $i$ at the $k$-th resolution level can be calculated using the entropy of the inferred label $\hat{y}_{k,i}$ as follows:

$$u_{k,i} = -\hat{y}_{k,i} \log \hat{y}_{k,i} - (1 - \hat{y}_{k,i}) \log(1 - \hat{y}_{k,i}) \tag{3}$$

We select a subset of cells with the highest uncertainty at each resolution level to refine the spatial partitioning. Let $T_k$ be a threshold for selecting high-uncertainty cells at the $k$-th resolution level. We define a set of cells $\{s_{k,i} \mid u_{k,i} \geq T_k\}$ that will be refined to the next finer resolution level $(k - 1)$.

For the selected cells in $\mathbf{S}_k$, we construct a new spatial partitioning with smaller cell size and update the grounding atoms set accordingly. We then perform PSL inference using the hierarchical label inference module at the $(k - 1)$-th resolution level with only the leaf node in the hierarchy. Since the distance-based loss is convex, we can use gradient descent to optimize it. To initialize $I$ at different resolutions, in the first iteration, *i.e.*, we pre-train the deep learning model with limited labels and use the output probabilities as the initialization. In the following iterations, the predicted probabilities of the corresponding deep learning model are regarded as the initial soft truth value of the ground atom in each $r$. Starting from the coarsest resolution $(k = K)$, the process continues iteratively until the finest resolution $(k = 0)$ is reached.

## 3.3 Uncertainty-aware deep learning

The uncertainty-aware deep learning module is capable of capturing information from the explanatory features and plays a significant role in handling the uncertainty of inferred labels and variations in resolution. In traditional deep learning models, the model makes a prediction for each sample, but it doesn't utilize any information about how confident the model is about that prediction [8]. This could lead to overconfident predictions in regions with scarce or noisy labels.

The module employs a modified version of the Binary Cross Entropy (BCE) loss to manage the uncertainty from the spatial knowledge and inferred label. In this module, all the deep learning

predictions are at the finest resolution. We replace the binary ground truth labels in the BCE loss function with the inferred uncertain labels. The adjusted cross entropy quantifies the difference between the predicted and inferred label probability distributions, effectively incorporating uncertainty into the training process and enhancing performance in ambiguous scenarios. The module addresses multi-instance learning scenarios encountered in partitioned spatial domains by computing an aggregate probability output for each pixel at various resolution levels, capturing the overall event likelihood within corresponding coarser cells. The loss function thus becomes:

$$L_{DL} = -\sum_{i=1}^{N_k} \hat{y}_{k,i} \log P_{k,i} + (1 - \hat{y}_{k,i}) \log(1 - P_{k,i}) \qquad (4)$$

The probability output $P_{k,i}$ for each cell sample $s_{k,i}$ is computed as:

$$P_{k,i} = \frac{1}{|s_{k,i}|} \sum_{s_{0,j} \in s_{k,i}} p_j \qquad (5)$$

where $\hat{y}_i \in \hat{\mathbf{Y}}$ is the inferred uncertain label, $P_{k,i}$ is the average of the predicted probabilities $p_j$ for all the samples $s_j$ within the coarse cell $s_{k,i}$. $|s_{k,i}|$ represents the number of finest resolution pixels in the cell.

This modification effectively incorporates uncertainty information into the training process and can improve the model's performance when dealing with ambiguous cases. What's more, it allows the model to handle different levels of granularity in the spatial domain, making it flexible and adaptable to various spatial scales.

## 4 Evaluation

### 4.1 Experiment setup

For the experiments, we use two real-world flood mapping datasets collected from North Carolina during Hurricane Matthew in 2016. We compare our proposed SKI-HL model with a variety of baselines that represent different approaches to handling spatial data and infusing knowledge into deep learning: **Pretrain**, **Self-training**, **DeepProbLog [18]**, **Abductive Learning (ABL) [3]**, and **SKI-HL-Base**. We used precision, recall, and F1 score on the flood mapping class to evaluate the pixel-level classification performance, and used $AvU_A$, $AvU_I$, and $AvU$ [16, 9] to evaluate the uncertainty quantification performance. The spatial knowledge base for the flood mapping task is based on distance and topology relationships [27, 14]. Please see Appendix C for more experiment details.

### 4.2 Comparison on classification performance

Table 2: Comparison on classification and uncertainty quantification for Dataset 1.

| Method | | Acc | | | | | Uncertainty | | |
|---|---|---|---|---|---|---|---|---|---|
| | Class | P | R | F1 | Avg. F1 | Acc | Accuracy | $AvU_A/AvU_I$ | $AvU$ |
| Pretrain | Dry | 0.79 | 0.62 | 0.70 | 0.74 | 0.75 | Accurate | 0.81 | 0.45 |
| | Flood | 0.73 | 0.86 | 0.79 | | | Inaccurate | 0.32 | |
| Self-training | Dry | 0.60 | 0.83 | 0.70 | 0.78 | 0.81 | Accurate | 0.85 | 0.65 |
| | Flood | 0.93 | 0.81 | 0.86 | | | Inaccurate | 0.53 | |
| DeepProbLog | Dry | 0.73 | 0.78 | 0.75 | 0.81 | 0.83 | Accurate | 0.90 | 0.40 |
| | Flood | 0.88 | 0.85 | 0.87 | | | Inaccurate | 0.26 | |
| ABL | Dry | 0.66 | 0.78 | 0.72 | 0.79 | 0.81 | Accurate | 0.85 | 0.44 |
| | Flood | 0.90 | 0.83 | 0.86 | | | Inaccurate | 0.29 | |
| SKI-HL-Base | Dry | 0.95 | 0.93 | 0.94 | **0.95** | **0.95** | Accurate | 0.82 | 0.69 |
| | Flood | 0.96 | 0.97 | 0.96 | | | Inaccurate | 0.59 | |
| SKI-HL | Dry | 0.96 | 0.92 | 0.94 | **0.95** | **0.95** | Accurate | 0.80 | **0.74** |
| | Flood | 0.95 | 0.98 | 0.96 | | | Inaccurate | 0.68 | |

We evaluated each model using 4 labeled pixels, with results in Table 2 highlighting SKI-HL's superiority over baselines. See Appendix D for the results on dataset 2. The Pretrain model struggles the most, likely due to surface obstacles disrupting classifier generalization. While Self-training

surpasses Pretrain, its predictions, though high confidence, can be erroneous, and its lack of spatial knowledge integration curtails performance. Both DeepProbLog and ABL underscore the value of integrating spatial knowledge. ABL, relying on first-order logic as rigid constraints for label revisions, falters with intricate spatial rules possessing inherent uncertainties. DeepProbLog, while promising, requires patch-level inference given its design constraints, impacting its efficacy. SKI-HL consistently tops the baseline models across datasets. Even without grounding every pixel, it matches its base model on Dataset 1, credited to its uncertainty-driven hierarchical label inference. This structure negates the need for full dense labeling, a hurdle for other models. Overall, SKI-HL epitomizes the merits of merging spatial domain knowledge with deep learning, particularly for expansive spatial tasks with scant training labels.

## 4.3 The effect of the number of initial labeled samples

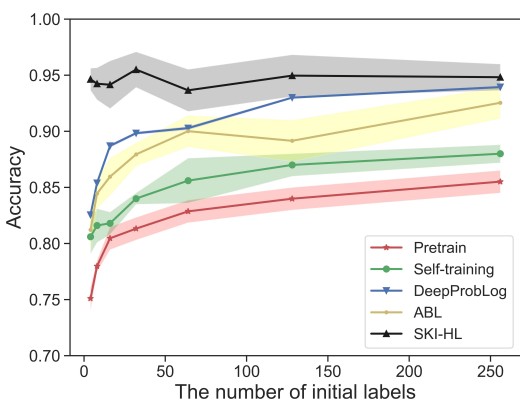

Figure 2: Accuracy comparison on different numbers of initial labels.

To rigorously assess our SKI-HL method, we tested Dataset 1 with initial labeled data ranging from 4 to 256, doubling at each step. We used the classification accuracy over 5 runs as the metric, presented in Figure 2. All baseline models improve with more labels. However, the gains in Pretrain and Self-training are modest due to their reliance on labeled pixels to represent entire patches. ABL, despite improvements, faces a performance ceiling because of its rigid logic for label revisions. DeepProbLog, using a logic-based framework for gradients, shows steady improvement with minimal variance. Distinctly, SKI-HL's accuracy remains consistent at around 0.95, regardless of the initial label count. This can be attributed to its unique label inference, which starts coarsely and refines iteratively. Minor result variations arise from training dynamics and initialization.

## 4.4 Comparison on uncertainty quantification performance

In Table 2, we notice a clear distinction in uncertainty estimation between our proposed SKI-HL model and the baselines. While Pretrain and Self-training models manifest a larger gap between $AvU_A$ and $AvU_I$, this discrepancy is mitigated in DeepProbLog and ABL, which effectively incorporate spatial knowledge into learning. However, they still struggle to achieve a balanced $AvU_A$ and $AvU_I$, particularly in situations of sparse and noisy labels. In stark contrast, our proposed SKI-HL model exhibits a superior performance on both datasets, signifying its robust ability to model complex spatial dependencies and adjust to areas of uncertainty dynamically. The integration of uncertainty-guided hierarchical label inference further mitigates the impact of sparse labeling, a bottleneck for other models. This finding emphasizes the pivotal role of efficiently integrating spatial domain knowledge with deep learning, especially under the constraints of limited training labels, in achieving reliable uncertainty estimation for large-scale spatial applications.

## 4.5 Case study

In our case study, we visually analyze the effectiveness of our model across varying resolution levels. As depicted in Figure 3a, we present the aerial Earth imagery, ground truth label, and digital elevation map from Dataset 1. It is noted that we don't use the ground truth to train our label, instead, it was

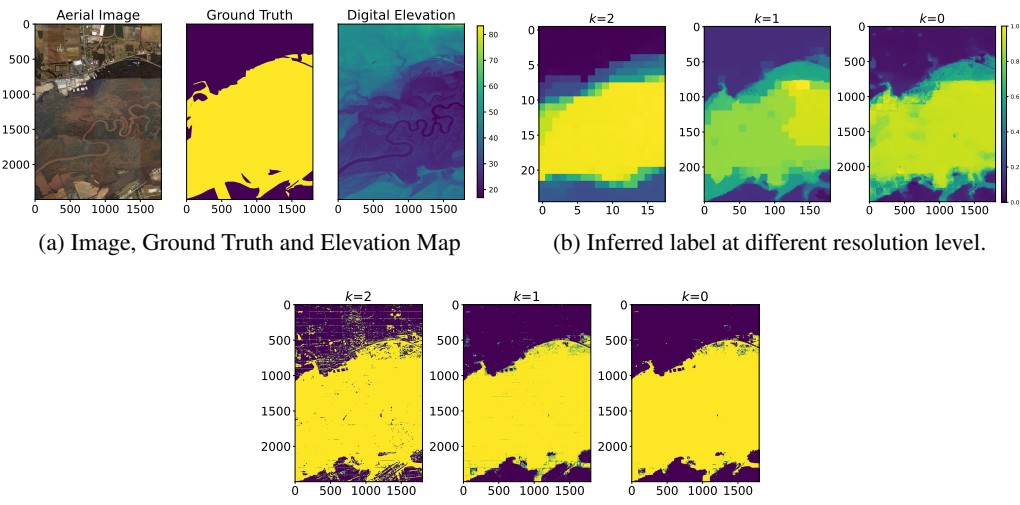

(a) Image, Ground Truth and Elevation Map    (b) Inferred label at different resolution level.

(c) Deep learning prediction at different resolution level.

Figure 3: Performance with different resolution in Dataset 1.

only used for testing. Figures 3b and 3c illustrate the evolution of inferred labels and deep learning predictions at different resolution levels. The resolution of the inferred labels refines progressively from a coarse resolution of 25 by 18 to the finest resolution of 2500 by 1800. This process allows for the accurate detection and refinement of uncertain areas, which often represent flood boundaries. Simultaneously, the granularity increase of the training labels results in an improved output from the deep learning model. A clear reduction in misclassified pixels can be observed, appearing as noise within each class of the area. This improvement can be attributed to the fact that multi-instance learning, used with coarse resolution labels, cannot provide supervision to every pixel. Hence, as our approach refines the label resolution, the deep learning model is able to generate more accurate predictions.

## 5    Conclusion and future works

In this paper, we proposed a novel Spatial Knowledge-Infused Hierarchical Learning (SKI-HL) framework that successfully addresses the limitations of existing deep learning models for Earth science through a system of iteratively inferring labels within a multi-resolution hierarchy. Our model outperformed several baseline methods on real-world flood mapping datasets.

In the future, the model can incorporate temporal dynamics features and capture changes in Earth imagery over time, which is critical for many applications such as deforestation tracking. Second, we can expand to other Earth science applications to further validate the generalizability and adaptability of the proposed SKI-HL framework.

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

# A  Preliminaries of symbolic logic

**Definition 1** *A **predicate** is a relation among objects in the domain or attributes of objects (e.g., Adjacent), and an **atom** is a predicate symbol applied to a tuple of terms (e.g., $Adjacent(s_i, s_j)$).*

**Definition 2** *A **rule** in logic is a clause recursively constructed from atoms using logical connectives and quantifiers. An example would be: $Flood(s_i) \wedge Adjacent(s_i, s_j) \rightarrow Flood(s_j)$.*

**Definition 3** *A **ground atom** $a$ and a **ground rule** $r$ are specific variable instantiations of an atom and rule, respectively. A **grounding** of an atom or rule is a replacement of all of its arguments by constants.*

**Definition 4** *The **t-norm fuzzy logic** can be used to relax binary truth value to continuous value between $[0, 1]$. The logical conjunction ($\wedge$), disjunction ($\vee$) and negation ($\neg$) are as follows:*

$$
\begin{aligned}
I(a_1 \wedge a_2) &= \max\{I(a_1) + I(a_2) - 1, 0\} \\
I(a_1 \vee a_2) &= \min\{I(a_1) + I(a_2), 1\} \\
I(\neg a_1) &= 1 - I(a_1)
\end{aligned}
\tag{6}
$$

# B  Spatial hierarchical structure

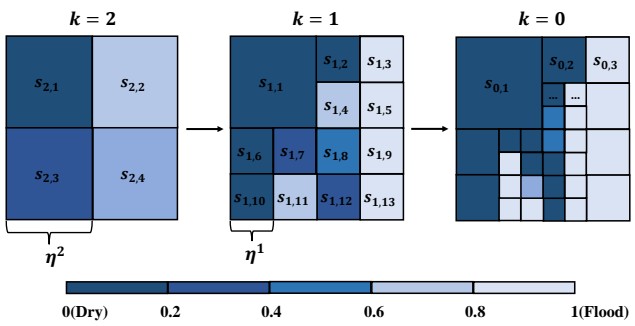

(a) An example for three resolution levels.

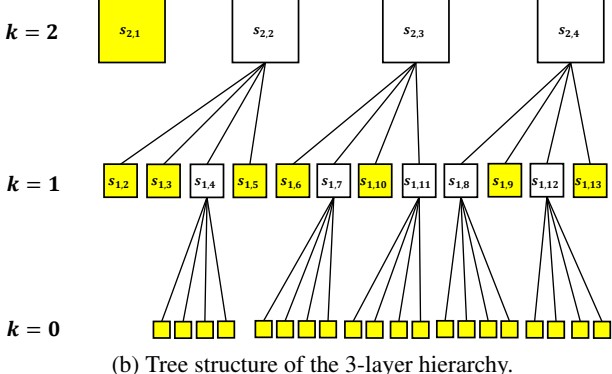

(b) Tree structure of the 3-layer hierarchy.

Figure 4: Illustration of hierarchical structure.

Given a large-scale spatial raster framework composed of pixels, we can represent the raster at multiple resolutions. At the coarse level, we can treat each cell as a condensed representation of many pixels. For example, in Figure 4a, if the original raster is $8 \times 8$ pixels (the rightmost grid), a coarser level representation (the leftmost grid) could be a $2 \times 2$ cell grid, where each cell represents a $2 \times 2$ pixel area of the original raster.

In the hierarchical structure of the spatial raster, the coarsest level grid represents the root of the hierarchy, and each subsequent refinement to a finer resolution represents a branching in the hierarchy.

As shown in Figure 4b, in this tree-like structure, each node represents a region with different sizes of the original spatial raster at the finest resolution. With the hierarchical structure, we can start the logic inference from coarse resolution, which decreases the proportion of unlabeled data and makes the inference practical. In the following, we provide a more formal description of the architecture.

Let $\mathbf{S}_0$ be an original large-scale spatial raster framework at the finest resolution. We define a constant $\eta$ where $\eta \in \mathbb{N}$ and $\eta > 1$, then we can get a set of resolutions $\{1, \eta, \eta^2 \ldots, \eta^K\}$ which denotes the grid sizes at $K+1$ levels. We use the power number to represent the indexes of the layers $k$ in the hierarchical structure, i.e., $k = 0, 1, 2, \ldots, K$. The set of samples in each layer $\mathbf{S}_k$, for $k \in \{1, \cdots, K\}$, represents the spatial raster at a specific resolution. The highest layer $K$ corresponds to the coarsest resolution and the lowest layer $0$ corresponds to the finest resolution, usually the original resolution. Each spatial sample $s_{k,i}$ at layer $k$ corresponds to a group of cells at the next finer layer $k-1$. To exemplify the hierarchy, consider Figure 4 representing 3 varying resolution levels. In this example, the grid size constant $\eta = 2$ and we have 3 layers in the hierarchy, i.e., $k = 0, 1, 2$. We select a subset of cells from layer 2 to "zoom in" further into smaller cells. This partitioning process results in layer 1, which contains both the original cells and the divided ones. The process is then repeated on the second grid, creating layer 0, which symbolizes the finest resolution layer. Through this hierarchical approach, each cell $s_{k,i}$ in layer $k$ corresponds to a group of cells in the next finer layer $k-1$, e.g., cell $s_{2,2}$ corresponds to cell $s_{1,2}, s_{1,3}, s_{1,4}$, and $s_{1,5}$ in Figure 4. At a certain level, we only need to ground the "leaf node", as shown in Figure 4b, which significantly decreases the number of ground atoms. Each layer in the hierarchy form a ground knowledge base $\mathcal{KB}_k = \{r_{k,1}, r_{k,2}, \cdots, r_{k,|\mathcal{KB}_k|}\}$ and then generates a set of inferred labels $\hat{\mathbf{Y}}_k = \{\hat{y}_{k,1}, \hat{y}_{k,2}, \cdots, \hat{y}_{k,N_k}\}$ and their corresponding uncertainties $\mathbf{U}_k = \{u_{k,1}, u_{k,2}, \cdots, u_{k,N_k}\}$, where $N_k$ is the number of samples, i.e., grid cells at the $k$-th resolution level.

To select the uncertainty area, taking layer 2 in Figure 4 as an example, each cell in this grid is color-coded to denote the probability of dry (dark) and water (light). We view the cell with the leftmost and rightmost color (certain Dry and Flood) in the color bar as certain cells, and others as uncertain cells, i.e., only the $s_{2,1}$ can be view as a certain cell in layer 2. For the selected cells in $\mathbf{S}_k$, we construct a new spatial partitioning with smaller cell size and update the grounding atoms set accordingly. As shown in Figure 4, the uncertain coarser cells in layer 2 are split into $2 \times 2$ finer cells, respectively.

## C Experiment setup details

**Dataset Description:** We use two real-world flood mapping datasets collected from North Carolina during Hurricane Matthew in 2016. The explanatory features comprise the red, green, and blue bands within the aerial imagery obtained from the National Oceanic and Atmospheric Administration's National Geodetic Survey[2]. In addition, digital elevation imagery was sourced from the University of North Carolina Libraries[3]. Each piece of data was subsequently resampled to a 2-meter by 2-meter resolution to standardize the information. For Dataset 1, the image has a shape of $2500 \times 1800$ with 4.5 million pixels. For Dataset 2, the image has a shape of $3400 \times 8400$ with 28.56 million pixels. In alignment with the principles of transductive learning, the experiment leverages both explanatory features and spatial information across the entire area throughout the learning process. A sparse set of labeled pixels forms the training set while the test set includes the whole area, excluding the labeled pixels.

**Candidate Methods:** In our experiments, we compare our proposed SKI-HL model with a variety of baselines that represent different approaches to handling spatial data and infusing knowledge into deep learning.

- **Pretrain:** In this method, the deep learning model is trained with the initially labeled pixels for each dataset.

- **Self-training:** The model adds patches with high confidence from Pretrain to the training dataset and re-trains the model.

---

[2]https://www.ngs.noaa.gov/
[3]https://www.lib.ncsu.edu/gis/elevation

- **DeepProbLog [18]:** This is a programming language that integrates deep learning with probabilistic logic programming. It allows for the incorporation of neural networks within a logic program, and these neural networks can be used to define probabilistic facts.
- **Abductive Learning (ABL) [3]:** This is a learning framework that combines both reasoning and learning. It works by training a model to make predictions and then using a logic reasoner to validate these predictions against a set of given logic rules. If the prediction contradicts the rules, the learning algorithm will revise its model based on the abductive explanation
- **SKI-HL-Base:** This is a simplified version of our model as a candidate method that doesn't implement the selection of uncertain areas in the grounding process. Instead, it ground all atoms for each layer in the hierarchy.

Table 3: Accuracy versus Uncertainty (AvU).

|  |  | Uncertainty | |
| --- | --- | --- | --- |
|  |  | Certain | Uncertain |
| Accuracy | Accurate | Accurate Certain (AC) | Accurate Uncertain (AU) |
|  | Inaccurate | Inaccurate Certain (IC) | Inaccurate Uncertain (IU) |

**Classification evaluation metrics:** We used precision, recall, and F1 score on the flood mapping class to evaluate the pixel-level classification performance.

**Uncertainty quantification evaluation metrics:** The performance of uncertainty estimations in our model is quantitatively evaluated using the Accuracy versus Uncertainty ($AvU$) measure, as shown in previous work [16, 9]. We set an uncertainty threshold, denoted by $T_u$, to group uncertainty estimations into 'certain' and 'uncertain' categories. Predictions based on these estimations are then grouped into four categories: Accurate-Certain (AC), Accurate-Uncertain (AU), Inaccurate-Certain (IC), and Inaccurate-Uncertain (IU). Let $n_{AC}, n_{AU}, n_{IC}, n_{IU}$ represent the number of samples in the respective categories. The $AvU$ measure evaluates the proportion of AC and IU samples, with the idea being that accurate predictions should ideally be accompanied by certainty, and inaccurate predictions should correspondingly indicate uncertainty. This measure lies in the range $[0, 1]$, with higher values indicating more reliable model performance. Specifically, we compute $AvU_A$ for accurate predictions and $AvU_I$ for inaccurate predictions as follows:

$$AvU_A = \frac{n_{AC}}{n_{AC} + n_{AU}}, AvU_I = \frac{n_{IU}}{n_{IC} + n_{IU}} \tag{7}$$

In our evaluation, we compute the harmonic average of $AvU_A$ and $AvU_I$ to penalize extreme cases:

$$AvU = \frac{2 * AvU_A * AvU_I}{AvU_A + AvU_I} \tag{8}$$

This evaluation approach thus offers a comprehensive measure of the reliability of our model's uncertainty estimations.

**Model configuration:** When implementing our method and baselines, we considered U-Net, a powerful deep learning model for image segmentation, as the base model. We set the same set of architecture for the U-Net model in all baselines with 5 downsample operations and 5 upsample operations. There is a batch normalization within each convolutional layer and the dropout rate is 0.2.

For Pretrain, images are divided into 100 by 100 patches, using patches containing a labeled pixel for training. All pixels in these patches are assigned the label of the initially labeled pixel for pre-training. Self-training uses Pretrain predictions to enhance the training dataset, iteratively adding high-confidence patches based on average predicted class probabilities. The same pretrained U-Net initializes the deep learning models in DeepProbLog, ABL, and our proposed SKI-HL frameworks. For DeepProbLog, the time cost of pixel-level inference is intractable, so we can only conduct patch-level inference.

For the hierarchical structure of SKI-HL, we set the grid size constant $\eta = 10$ and $K = 2$ which means there are 3 layers with grid size 100, 10, and 1 respectively in this hierarchy. We construct the spatial knowledge base for the flood mapping task based on distance and topology relationships. For the distance relationship, we directly use the neighborhood pair to model. For the elevation, we adopt a Hidden Markov Tree model [27, 14] which can model the topological relationship of each location based on the elevation.

# D Comparisons on dataset 2

Table 4: Comparison on classification and uncertainty quantification for Dataset 2.

| Method | | Acc | | | | | Uncertainty | | |
|---|---|---|---|---|---|---|---|---|---|
| | Class | P | R | F1 | Avg. F1 | Acc | Accuracy | $AvU_A/AvU_I$ | $AvU$ |
| Pretrain | Dry | 0.88 | 0.76 | 0.81 | 0.74 | 0.76 | Accurate | 0.97 | 0.20 |
| | Flood | 0.59 | 0.77 | 0.67 | | | Inaccurate | 0.11 | |
| Self-training | Dry | 0.89 | 0.77 | 0.83 | 0.76 | 0.78 | Accurate | 0.92 | 0.37 |
| | Flood | 0.61 | 0.80 | 0.69 | | | Inaccurate | 0.23 | |
| DeepProbLog | Dry | 0.83 | 0.87 | 0.85 | 0.82 | 0.82 | Accurate | 0.79 | 0.61 |
| | Flood | 0.82 | 0.77 | 0.79 | | | Inaccurate | 0.50 | |
| ABL | Dry | 0.76 | 0.86 | 0.81 | 0.79 | 0.79 | Accurate | 0.91 | 0.43 |
| | Flood | 0.82 | 0.71 | 0.76 | | | Inaccurate | 0.28 | |
| SKI-HL-Base | Dry | 0.92 | 0.94 | 0.93 | 0.91 | 0.91 | Accurate | 0.51 | 0.64 |
| | Flood | 0.91 | 0.88 | 0.90 | | | Inaccurate | 0.86 | |
| SKI-HL | Dry | 0.91 | 0.97 | 0.94 | **0.92** | **0.93** | Accurate | 0.80 | **0.84** |
| | Flood | 0.95 | 0.88 | 0.91 | | | Inaccurate | 0.88 | |

# E Analysis of time costs with hierarchical label inference

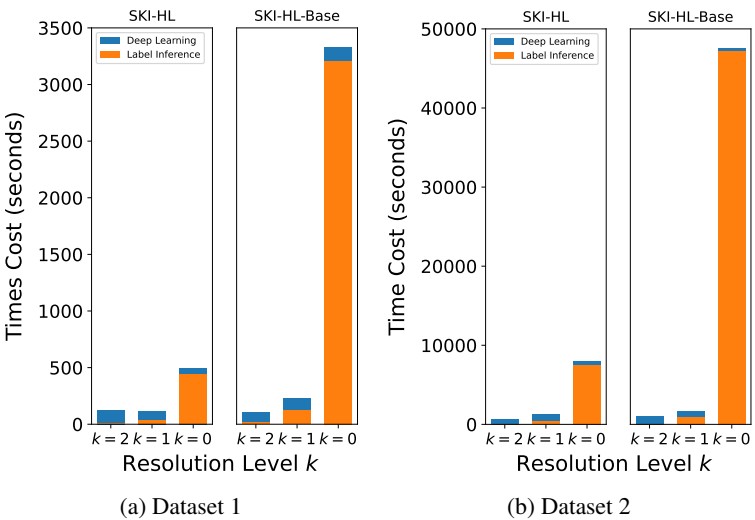

(a) Dataset 1        (b) Dataset 2

Figure 5: Comparison on time cost.

In order to demonstrate the computational efficiency of our proposed approach, we conducted a set of experiments evaluating the time costs of SKI-HL and its base model across different resolution levels. These experiments were executed on an AMD EPYC 7742 64-Core Processor CPU and an NVIDIA A100 GPU equipped with 80 GB of memory.

Figure 5 presents the time costs associated with the uncertainty-aware deep learning model training (blue bar) and the hierarchical label inference module training (orange bar) at each resolution level. Notably, the training time costs of the deep learning model remain relatively stable across different iterations, whereas the label inference process exhibits a strong dependency on the number of ground atoms. In particular, the label inference module requires the most significant computational resources when the number of ground atoms is large.

To illustrate, in dataset 2, the label inference at the finest resolution consumed approximately 12.5 hours without implementing a selective grounding process. However, when adopting the uncertainty-guided grounding strategy, the model achieved a considerable time-saving factor of 6.3 and 5.0 times compared to grounding all atoms for datasets 1 and 2, respectively. This stark difference in

computational time underlines the necessity and effectiveness of our proposed uncertainty-guided hierarchical label inference in the context of large-scale spatial data.

