# OpenReview forum: "Infusing Spatial Knowledge into Deep Learning for Earth Science: A Hydrological Application"
_NeurIPS.cc/2023/Workshop/AI4Science — NeurIPS2023-AI4Science Poster_

### Official Review · Reviewer_1eSt · 2023-10-17
**Overall good paper**

**Rating:** 7
**Confidence:** 4

**Review:**

The authors proposed a deep learning framework that integrates spatial knowledge. The model is proved to be effective with real hurricane datasets. It is well-organized and fit the workshop. I only have several comments.

comments:
In table 2, what is the difference between SKI-HL-Base and SKI-HL?
In Figure3, what is the unit for horizontal resolution (i.e., meters or kilometers)?
The U-net model is selected as the base model. Does it change the result if another base model is selected?

---

### Meta-Review · Area_Chair_GQ4c · 2023-10-27

**Recommendation:** Accept (Poster)
**Confidence:** 3

**Metareview:**

The authors introduce a deep learning framework that blends spatial understanding and uses an iterative training approach with labels from a multi-resolution hierarchy, factoring in uncertainty. The model's effectiveness is demonstrated on real-world hydrological datasets. I find this paper to be a significant contribution to the geospatial research community and recommend its acceptance for the workshop.